# Clinical Study of 14 Cases of Bone Augmentation with Selective Laser Melting Titanium Mesh Plates

**DOI:** 10.3390/ma16216842

**Published:** 2023-10-25

**Authors:** Ayaka Takahashi, Kazuya Inoue, Naoko Imagawa-Fujimura, Keisuke Matsumoto, Kazuto Yamada, Yasuhisa Sawai, Yoichiro Nakajima, Takamitsu Mano, Nahoko Kato-Kogoe, Takaaki Ueno

**Affiliations:** Department of Dentistry and Oral Surgery, Faculty of Medicine, Osaka Medical and Pharmaceutical University, 2-7 Daigakumachi, Takatsuki 569-8686, Osaka, Japankazuto.yamada@ompu.ac.jp (K.Y.); yasuhisa.sawai@ompu.ac.jp (Y.S.); takamitsu.mano@ompu.ac.jp (T.M.);

**Keywords:** dental implant, bone augmentation, SLM titanium mesh plate

## Abstract

Additive manufacturing techniques are being used in the medical field. Orthopedic hip prostheses and denture bases are designed and fabricated based on the patient’s computer-aided design (CAD) data. We attempted to incorporate this technique into dental implant bone augmentation. Surgical simulation was performed using patient data. Fourteen patients underwent bone augmentation using a selective laser melting (SLM) titanium mesh plate. The results showed no evidence of infection in any of the 14 patients. In 12 patients, only one fixation screw was used, and good results were obtained. The SLM titanium mesh plate was good adaptation in all cases, with bone occupancy greater than 90%. The average bone resorption of the marginal alveolar bone from the time of dental implant placement to the time of the superstructure placement was 0.69 ± 0.25 mm. Implant superstructures were placed in all cases, and bone augmentation with SLM titanium mesh plates was considered a useful technique.

## 1. Introduction

Bone augmentation must be performed when there is insufficient bone volume in the area where dental implants are to be placed [1]. Bone augmentation is often performed using autologous or artificial bone, with a nonabsorbable or absorbable membrane used to anchor the bone [2]. However, some bone defects make bone formation with artificial bones and membranes difficult. Bone augmentation with a nonabsorbable membrane requires intraoperative trimming of the conventional titanium mesh plate, which prolongs operative time.

Screws are also needed to secure the membrane. There is a risk of damaging the adjacent tooth root when fixing the screws. Decreasing the number of screws reduces the risk of damage. If the position of screw fixation can be determined in advance based on computed tomography (CT) data, safer surgery will likely be possible.

Patients who require bone augmentation often have jaw bones with complex bone defect morphologies. Bone augmentation is necessary to allow for the placement of implants in these bone configurations. The membrane must be trimmed and manipulated to achieve the ideal bone morphology, but if the surgeons are not skilled enough, they may not be able to obtain sufficient bone volume to place the implants [3].

Therefore, we have developed a method of fabricating a titanium mesh plate from the patient’s CT data and using it in surgery. This technique enables precise fitting of the bone defect and restoration of the bone morphology [4].

We believe that the risk of root damage is reduced by decreasing the number of screws and operative time. Based on the patient’s CT data, a surgical simulation is performed to measure the bone defect site, and a mesh plate is designed to fit the defect site. Based on the designed data, a titanium mesh plate is fabricated using the three-dimensional (3D) laminated molding technology, selective laser melting (SLM).

We have been performing bone augmentation using SLM titanium mesh plates, which can be freely formed based on computed tomography (CT) data, to improve alveolar bone conformity.

In this report, 14 cases of bone augmentation using SLM titanium mesh plates are described, and the usefulness of this technique is assessed.

## 2. Materials and Methods

### 2.1. Patients

Fourteen patients who underwent implant surgery at the Department of Dentistry and Oral Surgery, Faculty of Medicine, Osaka Medical and Pharmaceutical University from September 2016 to January 2022 were included. The Ethics Committee of Osaka Medical and Pharmaceutical University approved this study (approval no. 2311 and 2018-081-4). The 14 patients who underwent bone augmentation with the SLM method were included in the study. All patients were informed and consented to the treatment.

### 2.2. Fabricate SLM Titanium Mesh

The SLM titanium mesh plates were made by using surgical simulations based on Digital Imaging and Communications in Medicine (DICOM) data obtained after CT imaging [5]. The CT images of the patient for whom the titanium mesh plate was fabricated were taken, and metal artifacts were removed. Using the obtained DICOM data, a surgical simulation was performed based on the number of implants and their placement angles. The bone morphology and bone volume required for the implant placement site were then predicted, and a plate was designed using software (BioNa^®^ Wada Precision Dental Laboratories Co. Ltd., Osaka, Japan). The designed plate was converted into Standard Triangulated Language data on the computer. Data were then input into a metal 3D laminating machine (EOSINT M 270; EOS GmbH Electro Optical Systems, Krailling, Germany). A small hole was created in the SLM titanium mesh plate to ensure blood flow through the flap. Screw holes were also placed to fix the screws in advance. A patient-specific SLM titanium mesh plate with a thickness of 0.3 mm and screw hole diameter of 1.5 mm was fabricated using 3D laminated molding technology [4]. The surface was then polished using dental polishing point, and the plate was sterilized using a 121-degree autoclave and used in the surgery. The purpose of surface polishing was to remove the insoluble layer on the SLM titanium plate surface (Figure 1).

### 2.3. Evaluation Item

The usefulness of the SLM titanium mesh plate is discussed using 14 cases. The study aims were: (1) the number of screws used for fixation; (2) the presence or absence of mesh exposure and postoperative infection; (3) the presence or absence of simultaneous placement with bone augmentation; and (4) implant stability quotient (ISQ) values at the time of implant primary surgery and at the time of implant secondary surgery [6]. The ISQ is measured via the stimulation and resonance of a smart peg attached to an implant by magnetic pulses using the RFA (resonance frequency analysis) method. The frequency of the magnetic pulse at which resonance occurs correlates with the stability of the implant. ISQ values were measured using an Osstell IDx [7,8] Tooth Contact Analysis Device (Osstell AB, Gothenburg, Sweden). In this study, the period of unloading between the primary and secondary surgeries was 5–6 months for the maxilla and 2–3 months for the mandible, and the significance of differences in ISQ values was examined using a Mann–Whitney U test. (5) The hard tissue occupancy of the bone augmentation area was measured according to the method of Yamamoto et al. [9]. The region of interest (ROI), which is a range specification used for image processing of a specific area was defined as the bone augmentation area predicted from the morphology of the bone defect in an arbitrary cross-section on the preoperative CT images, and the average area occupied by the hard tissue in the cross-section divided into three equal sections from the center of the ROI on the postoperative CT images was used as the representative value of the volume increase and was measured as the hard tissue occupancy rate of the bone augmentation area. The bone resorption rate was measured. The height of the alveolar bone adjacent to the placed dental implants was measured using radiographs taken immediately after placement and at the time of superstructure implantation using the threads of the fixture as a marker. To ensure the accuracy of the measurements, measurements were taken twice by two people, two months apart. The average of the measured values was taken as the measured value.

### 2.4. Case Presentation

A 40-year-old man presented to our department with a maxillary incisor defect. He had undergone maxillary incisor tooth extraction at another dental clinic one year earlier. However, owing to the lack of bone mass, dental implant treatment was deemed impossible. The patient was referred to our clinic for further treatment. The patient’s medical history and drug history were unremarkable. Preoperative radiographs were obtained, and a surgical simulation was performed based on CT images. It was predicted that there would be insufficient bone mass on the labial side if implants were placed. A plan was made to perform primary implant surgery and bone augmentation simultaneously using an SLM titanium mesh plate. Surgery was performed under local anesthesia. After placement of a 3.5 mm diameter dental implant (SPI^®^ implant; Thommen Medical AG, Grenchen, Switzerland), bone loss on the labial side was observed. Bone augmentation was performed using an artificial bone (Bio-oss^®^ Geistlich Pharma, Wolhusen, Switzerland) and SLM titanium mesh plates. The ISQ value at primary surgery was 65. Six months later, secondary surgery was performed, and good bone augmentation was observed after removal of the SLM titanium mesh plate. The ISQ value at the time of the secondary surgery was 71. The patient’s postoperative course was uneventful, and he was fitted with a superstructure (Figure 2).

## 3. Results

(1)Number of screws used for fixation

The SLM titanium mesh was well adapted to the existing bone, and only one screw was used for fixation in 12 of the 14 cases. In the remaining two cases, a wide range of titanium mesh plates was used.

(2)Mesh exposure and infection

In one case, the SLM titanium mesh plate was slightly exposed; however, there was no evidence of infection. In the other 13 cases, implants were placed, and the upper prosthesis superstructure was installed.

(3)Simultaneous implant placement

Overall, 9 of the 14 patients underwent simultaneous implant placement and bone augmentation. The remaining five patients underwent implant placement 3–6 months after bone creation (Table 1).

(4)ISQ value

ISQ values were obtained for 25 implants. The ISQ was 67.98 ± 7.95 at primary surgery and 73.22 ± 6.70 at secondary surgery, showing a marked increase in the ISQ value at secondary surgery. The ISQ value was significantly improved in the secondary surgery compared to the primary surgery (Figure 3).

(5)Hard tissue occupancy of the bone augmentation area

Bone occupancy in the ROI averaged 88.4 ± 3.86%. No abnormal bone resorption was observed in any area.

(6)Bone resorption rate

The average bone resorption of the marginal alveolar bone from the time of the dental implant placement to the time of the superstructure placement was 0.69 ± 0.25 mm. No areas of abnormal bone resorption greater than 5 mm were observed.

## 4. Discussion

One of the advantages of using additive manufacturing technology in the medical field is that it offers a high degree of freedom in terms of shape assignment, and advances in CAD/CAM technology have made it possible to freely create devices based on STL data converted from DICOM data of CT images. These technologies have made it technically possible to fabricate a fully custom titanium device that precisely fits an individual patient’s jaw configuration by designing the fixation plate from the patient’s CT images and laminating titanium powder. In the medical field, artificial hip and knee joints in the orthopedic field and custom-made mandible reconstruction plates in the oral surgery field have already been used. Additive manufacturing using pure titanium powder equivalent to grade two can provide the strength acceptable in vivo.

Bone augmentation is a procedure that aims to increase bone mass using artificial or autologous bone grafts to compensate for bone loss in areas where the jaw bone has undergone vertical or horizontal bone resorption and implants cannot be inserted [1,10]. A shielding membrane is fixed in the void space of the artificial bone or autogenous bone graft. Shielding membranes can be classified into two categories: absorbable and nonabsorbable. However, the process of applying and adapting nonabsorbable titanium mesh to the existing bone depends on the skill and experience of the surgeon. A lack of experience may slow the process and increase operative time, whereas the use of SLM titanium mesh may help solve the clinical problems associated with conventional titanium mesh sheets [11]. Conventional titanium mesh sheets are trimmed according to the morphology of the bone defect. However, the sharp edges of the trimmed titanium mesh sheet can lead to wound rupture, infection, and loss of the grafted bone; no cases of infection or wound rupture have been reported with the SLM titanium mesh plate because of its excellent fit [12].

The case of a patient with a maxillary anterior tooth was presented. SLM titanium mesh is very effective for use in the anterior region. The single tooth defects in the maxillary anterior region are often associated with a lack of horizontal bone volume. In addition, bone loss in a concave shape is observed. If implants are placed without osteogenesis, it is not possible to secure 1.5 mm of bone volume from the labial bone. Conventional titanium honeycomb membranes are used for osteogenesis, but the small space between adjacent teeth makes it difficult to determine the form of the bone augmentation and the position of screw fixation. The SLM titanium mesh plate is designed with a three-dimensional simulation of the screw fixation position, allowing for secure fixation.

SLM titanium mesh is also used in cases of vertical bone loss. In mandibular molars, bone resorption after extraction results in shortening of the distance between the bone margin and the inferior alveolar nerve, leading to vertical bone deficiency in some cases. In recent years, short implants [13,14] have come to be used due to improvements in implant surface modification technology [14,15]. However, short implants are at high risk of infection and implant loss due to gingival recession when the rough surfaces are exposed. The lack of long-term data undermines the reliability of this method. When vertical bone augmentation is used for mandibular molars, flap control is important. Even in our case with a molar, the SLM titanium mesh was slightly exposed. A good outcome could be achieved by performing a two-term free gingival graft.

Conventional bone augmentation using a titanium mesh often requires multiple screws for fixation because of insufficient conformity to the existing bone. Alessandro [16] evaluated the usefulness of bone grafting methods using canine jaws. Titanium-reinforced membranes and autogenous bone blocks showed significantly greater volume stability than GBR (guided bone regeneration) with a collagen membrane, especially in the coronal portion. GBR with additional membrane fixation showed better results than GBR without fixation.

The fixation screws must avoid the root of the tooth in the area to be implanted, and the risk of root damage increases when a large number of screws need to be implanted. Sumida et al. [17] reported that 3.23 ± 0.73 screws were used in the case of conventional mesh sheets during bone augmentation, but SLM titanium mesh plates used only one screw in most cases. Sumida also reported that the method using SLM titanium mesh saved time compared to the conventional method. In most cases, only one screw was required for fixation, thus reducing the risk of root damage.

There are some caveats to using SLM titanium mesh plates. When using a single nonlocking screw, the SLM titanium mesh plate rotates clockwise when strongly tightened during fixation. To prevent this, it is important that the assistant holds the plate in the correct position during fixation; the SLM titanium mesh plate may be useful for both vertical defects in molars and buccolingual defects in the anterior teeth.

The ISQ value is a quantification of the stability of an implant using a magnetic resonance device. An ISQ value of approximately 70 is considered to indicate adequate fixation [7], the authors used the ISQ to measure changes in the ISQ during primary and secondary implant surgery. They reported an average ISQ value of 66.3 ± 11.0 at primary surgery and 72.2 ± 6.7 at secondary surgery. In the present study, the ISQ value at primary surgery was 67.98 ± 7.95, which was lower than that reported by Matsumoto et al. This was the ISQ value at the bone augmentation site. The difference in ISQ values may be due to the lack of sufficient buccal or vertical bone volume at the time of primary implantation. The average ISQ value at the time of secondary surgery was 73.22 ± 6.70 and showed similar results. Vianna [18] reported that the amount of bone loss in the marginal alveolar bone after the placement of conventional dental implants was 0.5–0.9 mm. In this case, the amount of bone loss was about 0.69 mm, indicating that there was no difference from the amount of bone loss after the placement of a general dental implant body. Therefore, bone augmentation using SLM titanium mesh plates is considered to be an effective method.

One of the issues with additive manufacturing techniques is the need for surface polishing or chemical surface treatment after fabrication due to residual stress caused by incomplete melting of the surface and the removal of residual powder during fabrication [19]. However, the SLM titanium mesh plate did not result in any significant clinical complications. A previous study compared computer-aided design (CAD) design with an actual SLM titanium mesh plate, and the authors concluded that the reproducibility was sufficiently precise to meet ISO standards [20]. In the dental field, accuracy of less than 100 μm is required for crown prosthetics and about 300 μm for complete dentures, and it has been reported that the fit of devices fabricated by laser additive manufacturing technology is comparable to that of dental casting technology. Metal laminated molding technology using CAD is being used to fabricate denture frames for partial dentures and metal crowns [21,22]. Additive manufacturing technology is expected to continue to have a significant impact on the development of dental fields in the future.

## 5. Conclusions

A total of 14 cases of bone augmentation using an SLM titanium mesh plate were reported. Implant stability was elevated. No abnormal bone resorption was observed and sufficient bone occupancy was indicated. In all cases, a good clinical result followed. Bone augmentation with SLM titanium mesh plates was considered to be a useful method.

## Figures and Tables

**Figure 1 materials-16-06842-f001:**
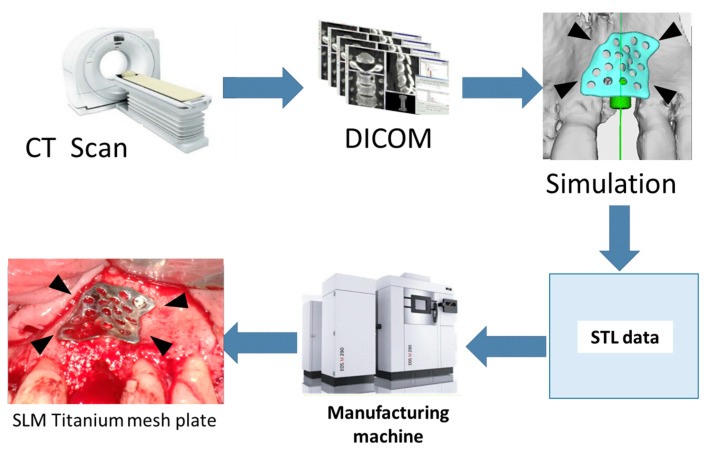
SLM titanium mesh plate protocol. Measurement of the bone defect is performed using three-dimensional measurement software; an SLM titanium mesh plate with dimensions appropriate for the necessary bone mass is designed; and the corresponding STL data are forwarded to a three-dimensional construction device. A 0.3 mm thick SLM titanium mesh sheet is manufactured with 1.5 mm diameter screw holes.

**Figure 2 materials-16-06842-f002:**
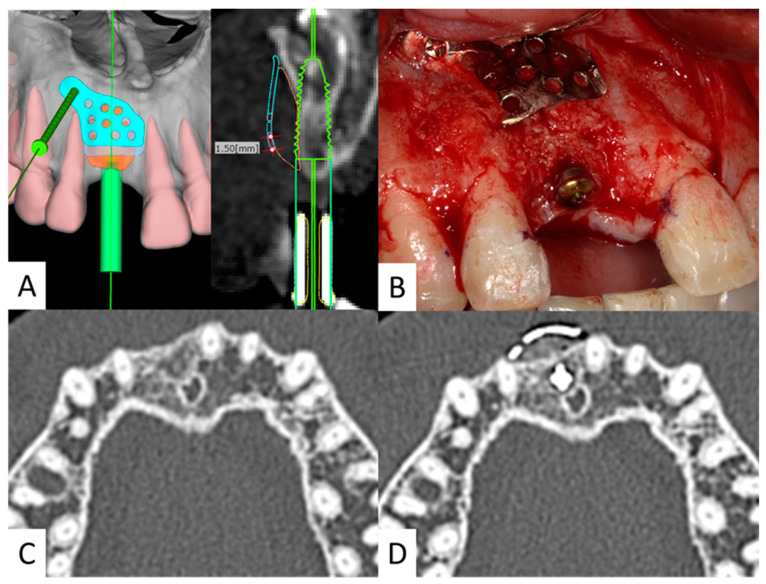
(**A**). CT preoperative simulation. Simulated surgical images. Preoperative simulation is performed to predict the bone defects and design the SLM titanium mesh plate. (**B**). Intraoperative photograph. Artificial bone and SLM titanium mesh plate are placed in the bone defect on the labial side. Only one screw is used for fixation. (**C**). Preoperative CT image. There is insufficient bone for implant placement. (**D**). Postoperative CT image. Sufficient bone volume is obtained on the labial side.

**Figure 3 materials-16-06842-f003:**
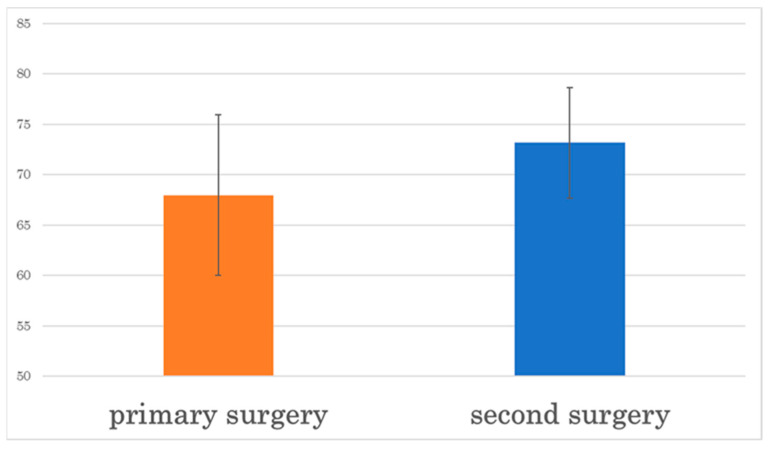
The ISQ value is significantly higher at secondary surgery than at primary surgery.

**Table 1 materials-16-06842-t001:** Summary of 14 cases of bone augmentation.

Case	Sex	Age	Missing Teeth	Screw	Implants	Bone Occupacy	Bone Resorption
1	F	82	43, 44	1	2	89.2	0.51
2	F	63	35, 36	1	2	84.8	0.33
3	M	73	45, 46	1	2	92.1	0.73
			35, 35	1	2	85.0	1.11
4	M	72	24, 25, 26, 27	1	3	87.0	1.12
5	F	56	21	1	1	88.0	0.93
6	F	71	11	1	1	92.0	0.87
7	M	45	22	1	1	96.7	0.52
8	M	55	11, 12, 13, 14	2	3	89.0	0.53
9	M	61	12	1	1	92.0	0.33
10	F	64	21, 23	1	1	93.0	0.35
11	M	55	14, 15	2	2	85.2	0.76
12	M	53	11, 12	1	2	83.0	0.65
13	F	72	25	1	1	87.0	0.96
14	M	46	11	1	1	83.3	0.68

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
