# Peer review of "Clinical Study of 14 Cases of Bone Augmentation with Selective Laser Melting Titanium Mesh Plates"

_materials, 2023, doi:10.3390/ma16216842_

Round 1
Reviewer 1 Report
As a case series report, 14 is a good number. The general structure of the report is good, but it needs some adjustments.
1. The introduction is poor in References: only one... It must be revised and some actual references must be added.
2. The paragraph of lines 79 to 83 is part of the introduction, not Materials and Methods
3. The reference to Figure 1 is missing in the text. The thickness of the SLM titanium in the legend of this figure is wrong: 300 micrometers, not 30.
4. A 2.2. section must be added to Material and Methods.
5. In Table 1, use FDI numbering method to describe missing teeth; The ISQ value is primary or secondary?; It would be nice to have an additional column with the indication of the simultaneous implant placement and the case of SLM titanium mesh exposed.
6. The lines 155-173 are introduction. They don't analyze the results nor compare them to other studies.
7. Line 182 and 183: "This can cause problems such as peri-implantitis" Remove this sentence. The facts reported aren't associated with peri-implantitis.
8. The reference in line 207 is wrong. Instead of 5 it should be 15.
9. In a total of 23 references, 6 (25%) are self-citations... It's too much. Some are clearly an abuse, as the 18.
10. Conclusions are too short. Information should be added to the evaluation of the 14 cases.
Author Response
Dear Authors,
Thank you very much for providing important insights. We are delighted to hear that you think our work will spark debate in our field. In the following sections, you will find our responses to each of your points and suggestions. We are grateful for the time and energy you expended on our behalf.
- The introduction is poor in References: only one... It must be revised and some actual references must be added.
Thank you for pointing this out. We added references.
- The paragraph of lines 79 to 83 is part of the introduction, not Materials and Methods
Thank you for pointing this out. This sentence was added to the introduction
- The reference to Figure 1 is missing in the text. The thickness of the SLM titanium in the legend of this figure is wrong: 300 micrometers, not 30.
Thank you for pointing this out. We corrected and unified the notation to 0.3mm.
- A 2.2. section must be added to Material and Methods.
Thank you for pointing this out. We added sections 2.1 through 2.4 to Material and Methods.
- in Table 1, use FDI numbering method to describe missing teeth; The ISQ value is primary or secondary?It would be nice to have an additional column with the indication of the simultaneous implant placement and the case of SLM titanium mesh exposed.
Thank you for pointing this out. We have changed the table as you indicated.
- The lines 155-173 are introduction. They don't analyze the results nor compare them to other studies.
Thank you for pointing this out. We have added this section to the Introduction.
- Line 182 and 183: "This can cause problems such as peri-implantitis" Remove this sentence. The facts reported aren't associated with peri-implantitis.
Thank you for pointing this out. We have reviewed it and removed this sentence.
- The reference in line 207 is wrong. Instead of 5 it should be 15.A
Thank you for pointing that out. We have renumbered the references.
- In a total of 23 references, 6 (25%) are self-citations... It's too much. Some are clearly an abuse, as the 18.
Thank you for pointing this out. We have changed to a different reference.
- Conclusions are too short. Information should be added to the evaluation of the 14 cases.
Thank you for pointing that out. I have added the conclusion.
Again, we appreciate all of your insightful comments. We worked hard to be responsive to them. Thank you for taking the time and energy to help us improve the paper.

Reviewer 2 Report
Dear Authors,
Thank you for submitting your manuscript detailing the use of selective laser melting (SLM) titanium mesh plates in bone augmentation for dental implant surgeries. The topic is undeniably pioneering and holds significant potential for advancing personalized medical interventions. While the manuscript demonstrates valuable insights, there are certain key areas that need revisiting to ensure it meets the highest scientific standards and provides unambiguous clarity for readers.
1. Sample Size: Your study incorporated 14 patients, but the rationale behind this specific sample size is not elucidated. Is this number statistically significant to derive the presented conclusions? Clarifying the methodology or logic behind selecting this sample size would bolster the manuscript's validity.
2. Control Group: The absence of a control group (e.g., patients undergoing traditional bone augmentation methods) is noteworthy. A direct comparison using a control group would lend more weight to the benefits and efficiency of the SLM titanium mesh plates.
3. Ethical Considerations: Although the manuscript alludes to informed consent from all patients and approval from an Ethics Committee, it would enhance the manuscript's integrity to dedicate a distinct section elaborating on these ethical aspects.
4. Statistical Analysis: Your choice of the Student’s t-test to compare pre- and post-surgery ISQ values raises questions regarding its applicability. Have the assumptions for the t-test (like data normality) been verified? Shedding more light on this, or considering alternative statistical tests, would solidify the results' credibility.
5. Terminology: The manuscript occasionally resorts to specialized terminologies, including but not limited to "ROI" and "ISQ". To cater to a broader readership, potentially inclusive of clinicians unfamiliar with these acronyms, it might be prudent to offer more comprehensive explanations or even append a glossary.
6. Limitations: While the manuscript does touch upon some limitations, like the necessity for surface polishing post-fabrication, a more consolidated and expansive section addressing the potential drawbacks or challenges of the discussed technology would provide a well-rounded perspective to readers.
Given the above, my recommendation leans towards accepting the manuscript once these major revisions have been implemented. Addressing the outlined points will undoubtedly enhance its value to those intrigued by the intersections of dental implant surgery and avant-garde manufacturing techniques.
Author Response
Dear Authors,
Thank you very much for providing important insights. We are delighted to hear that you think our work will spark debate in our field. In the following sections, you will find our responses to each of your points and suggestions. We are grateful for the time and energy you expended on our behalf.
- Sample Size: Your study incorporated 14 patients, but the rationale behind this specific sample size is not elucidated. Is this number statistically significant to derive the presented conclusions? Clarifying the methodology or logic behind selecting this sample size would bolster the manuscript's validity.
We wish to thank the reviewer for this comment. The range was calculated by multiplying the standard error by 1.96 to obtain a 95% confidence interval. clinically acceptable values for ISQ, bone occupancy, and bone resorption rate were set for each. We determined that approximately 10 cases were statistically valid.
- Control Group: The absence of a control group (e.g., patients undergoing traditional bone augmentation methods) is noteworthy. A direct comparison using a control group would lend more weight to the benefits and efficiency of the SLM titanium mesh plates.
We appreciate the reviewer's comment on this point. We also considered comparing them to a control group. However, we considered that the implant sites, age, and bone volume of the control group were different from those of the control group, and therefore, we did not consider them to be sufficiently comparable.
- Ethical Considerations: Although the manuscript alludes to informed consent from all patients and approval from an Ethics Committee, it would enhance the manuscript's integrity to dedicate a distinct section elaborating on these ethical aspects.
In accordance with the reviewer's comment, we have changed the sentence of material and method. We added the sentence [All patients were informed and consented to the treatment.]
- Statistical Analysis: Your choice of the Student’s t-test to compare pre- and post-surgery ISQ values raises questions regarding its applicability. Have the assumptions for the t-test (like data normality) been verified? Shedding more light on this, or considering alternative statistical tests, would solidify the results' credibility.
The reviewer's comment is correct.  Student-t was the incorrect statistical method. The correct statistical method should have been Man-whitney U because the data were not normally distributed.
- Terminology: The manuscript occasionally resorts to specialized terminologies, including but not limited to "ROI" and "ISQ". To cater to a broader readership, potentially inclusive of clinicians unfamiliar with these acronyms, it might be prudent to offer more comprehensive explanations or even append a glossary.
We agree that this point requires clarification, and have added the sentence to explain the ROI and ISQ.
- Limitations: While the manuscript does touch upon some limitations, like the necessity for surface polishing post-fabrication, a more consolidated and expansive section addressing the potential drawbacks or challenges of the discussed technology would provide a well-rounded perspective to readers.
We agree that this point requires clarification, and have added the sentence to explain the purpose of the polishing post-fabrication.
Again, we appreciate all of your insightful comments. We worked hard to be responsive to them. Thank you for taking the time and energy to help us improve the paper.

Round 2
Reviewer 1 Report
After the modifications, the quality of the manuscript improved.
Reviewer 2 Report
Dear Authors,
I appreciate your commitment to enhancing the quality of their work, and I believe the manuscript will contribute valuable insights to the field of Implantology and bone augmentation. I recommend its publication in its current form and look forward to seeing it shared with the wider academic community.
Thank you for considering my assessment, and please do not hesitate to contact me if further clarification or information is needed.